# Talking to Dogs: Companion Animal-Directed Speech in a Stress Test

**DOI:** 10.3390/ani9070417

**Published:** 2019-07-04

**Authors:** Raffaela Lesch, Kurt Kotrschal, Iris Schöberl, Andrea Beetz, Judith Solomon, W. Tecumseh Fitch

**Affiliations:** 1Department of Cognitive Biology, University of Vienna, Althanstrasse 14, 1090 Vienna, Austria; 2Department of Behavioral Biology, Konrad Lorenz Research Station, University of Vienna, Althanstrasse 14, 1090 Vienna, Austria; 3Department of Special Education, Institut für Sonderpädagogische Entwicklungsförderung und Rehabilitation, University of Rostock, August-Bebel-Str. 28, 18055 Rostock, Germany

**Keywords:** companion animal-directed speech, dog-directed speech, attachment, caregiving, Ainsworth strange situation, pet-directed speech

## Abstract

**Simple Summary:**

Companion animal-directed speech is a current topic of research, interesting due to its similarity to infant-directed speech. Dog owners seem to almost subconsciously use this high-pitched and repetitive way of speaking, slightly adapted, for dogs. The aim of this study was to investigate dog-directed speech in different contexts and examine whether owner personality and relationship quality affect it. We found that owners’ personality and gender affect their dog-directed speech. The majority of the modifications of dog-directed speech could be explained by a differential use of voice pitch and range. Our study supports the idea that voice pitch was used to communicate affect, whereas pitch range was used as an attention-getting strategy. Based on our results, we conclude that dog-directed speech is adjusted depending on context, gender, and personality. Societal value in this study consists of its contribution to basic knowledge of how we talk to animals, which may help in preventing accidents (e.g., dog bites) as well as improving animal training.

**Abstract:**

Companion animal-directed speech (CADS) has previously been investigated in comparison to infant-directed speech and adult-directed speech. To investigate the influence of owner caregiving, attachment pattern, and personality on CADS, we used the Ainsworth strange situation procedure. It allowed us to assess voice source parameters of CADS across different contexts. We extracted speech parameters (voicing duration, voice pitch, pitch range, and jitter) from 53 dog owners recorded during the procedure. We found that owner personality and gender but not caregiving/attachment behavior affect their voice’s pitch, range, and jitter during CADS. Further, we found a differential and context-specific modification of pitch and range, consistent with the idea that pitch communicates affect, whereas range is more of an attention-getting device. This differential usage, and the increased pitch, emphasize and support the parallels described between CADS and infant-directed speech. For the first time, we also show the effect of personality on CADS and lay the basis for including jitter as a potentially useful measure in CADS.

## 1. Introduction

It has long been known that the human voice carries substantial cues to the emotional state of the owner, completely independent of any verbal or linguistic content [1]. With the exception of a few familiar words (e.g., their name, “good”, “bad”, etc.), animals listening to human speech presumably rely primarily upon such nonverbal information to interpret speaker state and intentions [2]. However, the specific forms of information available in companion animal-directed speech (CADS) remain understudied. In this study, we analyzed dog-directed speech in different situations to gain insight into this question. We focused attention on source characteristics, particularly voice fundamental frequency (f0, often colloquially termed “voice pitch”), which has reliably been found to vary with emotional state, though often in individual-specific manners [3,4]. Based on this previous research, we hypothesized that voice pitch variables would increase in increasingly arousing situations. In addition to summary statistics on f0 itself (mean, range, duration), we also analyzed voice perturbation using jitter, which quantifies local period-to-period deviation in the length of successive f0 periods. Measures of voice pitch irregularities (jitter) have a long history in voice emotion analysis, and high jitter has been suggested to correlate with high stress or arousal [3,5]. However, inconsistencies across studies have been argued to reflect individual personality differences [1] and may vary with coping style (e.g., for inhibited vs. outgoing children [6]). We therefore had only a weak a priori prediction that jitter should increase with arousal, perhaps in individual- or personality-specific ways.

Research trying to identify information encoded in CADS has used infant-directed speech (IDS) as a comparison. When compared to ordinary speech, CADS has a higher fundamental frequency, higher pitch range, and is more repetitive; these characteristics have also been found in IDS [7,8,9]. CADS may be similar to IDS in quality but differs from IDS in an apparent lack of hyper-articulation and lack of a language learning context [7,10,11]. These similarities and subtle differences make the comparison all the more intriguing and fascinating. Not only do people use CADS with dogs, but dogs also seem to pay attention to it. CADS has been evaluated for its attention-getting quality with adult dogs and seems to draw adult dogs’ attention more than adult-directed speech [12]. It appears that we adapt CADS (e.g., omitting hyper articulation), knowing that our four-legged friends will never be able to speak, but we still use it in navigating our relationship with them. One of the most important functions within relationships is giving and receiving emotional social support. How we give and receive this emotional and social support depends on and varies with arousal level and attachment representations [13,14]. Humans biophilic nature combined with dogs’ adaptation to humans (domestication) allows both sides to use and respond to the aforementioned human offspring–caregiver attachment behavior [15,16].

Any attachment relationship is characterized by maintaining proximity, separation distress, and using the caregiver as a secure base to explore from, and as a safe haven to return to [17,18]. Mary Ainsworth developed a classification tool to diagnose the quality of an infant’s attachment [19,20], the Ainsworth strange situation procedure (ASSP). The ASSP is designed to assess attachment patterns by increasing the subjects’ mental stress load and arousal over consecutive episodes [21,22]. It is created to increasingly activate attachment behavior on the side of the child or dog; this behavior is usually reciprocated by the caregiver in a complementary manner [23], which means the experience of mental stress has the subject turn towards support and comfort, which is found in the form of social support [19]. This is why we want to investigate CADS in this setup: Social communication has a highly relevant function in providing emotional and social support in any relationship. Both attachment theory and the ASSP have previously been applied to human–dog dyads, with promising results [24,25,26,27]. Since literature suggests that CADS can be influenced by owner gender and context [28,29], the ASSP is an excellent tool for testing CADS in varying contexts. To date, analyses of CADS did not take into account important factors in social communication, such as attachment pattern, caregiving behavior or personality. Here, we want to investigate their potential impact on the voice source during CADS.

## 2. Materials and Methods

### 2.1. Subjects 

A total of 59 human–dog teams participated in this study. Six out of the 59 teams had to be excluded from the analysis due to recording equipment malfunctions. The remaining 53 human–dog teams (28 women, 25 men, 28 female dogs, 25 male dogs) in balanced combinations of male–male, female–female, male–female, and female–male were a subset of 132 human–dog dyads who had already participated in an experimental study of interaction styles and human–dog relationships [30,31]. The recruitment for our study was based on voluntary participation of teams from the pool of 132 dyads. Our subset of dyads represents those who responded positively to our request for participation in this study. All dogs lived with the owner who was also the main attachment figure from puppyhood onward. All dogs were intact, i.e., were neither spayed nor neutered. Their mean age was 4 years ± 1.5 SD and their mean weight was 30 kg ± 13.2 SD. Mean owner age was 46.2 years ± 10.2 SD. All dyads were recruited from Vienna and surrounding areas in Austria.

### 2.2. Owner Personality Axis

Dog owners were asked to fill in the German version of the NEO Five-Factory Inventory evaluating their own personality [32,33], a 60-item psychometric instrument designed to evaluate nonclinical adult personality structures along five major dimensions: neuroticism, extraversion, openness, agreeableness, and conscientiousness.

### 2.3. Ainsworth Strange Situation Procedure

The Ainsworth strange situation procedure [20] was adapted to assess the human–dog relationship. Within this assessment, dog attachment behavior is activated via increasing stress during the procedure, which in turn potentially activates caregiving behavior in the owner. Before testing, the room was prepared by placing two color-tagged chairs next to each other in the middle of the room, closing the windows and shades, and depositing several toys between and in front of the chairs. The dog owners were guided through the procedure by the experimenter in the following fixed order:

**0. Controls (~10 s):** Two control settings (reading and speaking/adult-directed speech) are recorded in the waiting room, adjacent the experimental room: First, the owner is asked to read a predefined text in presence of the dog out loud. (“…der beste Freund des Schäferhundes Rex ist die Ente Oskar…”—“… the German shepherd Rex’s best friend is the duck Oscar…”). For the second control, the experimenter engages the owner in small talk about their dog.

**1. Introduction (~20 s):** The experimenter leads the owner–dog dyad into the room, introduces them to the surroundings, asks the owner to unleash the dog for the duration of the procedure, and leaves the room again.

**2. Exploration (3 min):** Both dog and owner can move around freely and explore the room and the toys. The owner is free to interact and talk with the dog normally.

**3. Encounter (1 + 2 min):** A person previously unknown to the dog (‘the stranger’) enters the room and asks the owner to take their designated seat if they were not sitting yet. Other than this, there is no further interaction initially: For one minute, the stranger sits down on the other chair, remaining motionless. In the second minute, the stranger initiates small talk with the dog owner. In the third and last minute of this first encounter, the stranger tries to interact with and elicit play from the dog. At the end of minute three, the stranger asks the dog owner to leave the room.

**4. Separation (3 min):** The dog stays alone in the experimental room with the stranger. The stranger tries to engage the dog by playing and interacting.

**5. Call (~5 s):** The experimenter instructs the dog owner to stand in front of the experimental room door and to loudly call the dog’s name.

**6. Reunion (3 min):** The owner re-enters the room and reunites with the dog. The owner is free to interact and to play with the dog. The stranger quietly leaves the room. At the end of minute three, a vibrating phone indicates that the owner should leave the experimental room.

**7. Separation (3 min):** The dog remains alone in the experimental room for three minutes.

**8. Encounter (3 min):** The stranger from stage 3 re-enters and the dog remains in the experimental room with the stranger. The stranger tries to comfort the dog by playing and interacting.

**9. Call (~5 s):** The experimenter again instructs the dog owner to stand in front of the experimental room door and loudly call the dog’s name.

**10. Reunion (3 min):** The owner enters the room, reunites with the dog, and provides at least one physical contact. The owner is free to interact and play with the dog. The stranger quietly leaves the room. At the end of minute three, the experimenter enters the room and ends the procedure.

The procedure is illustrated graphically in Figure 1, marking all episodes (0, 2, 3, 5, 6, 9, 10) where the owner’s utterances were analyzed with the owner icon.

### 2.4. Video Recording, Surveillance and Animal Welfare

The test room was equipped with a wide-angle lens video surveillance system (Canon Inc., Canon Austria GmbH, 1100 Vienna, Austria) to record and monitor the proceedings throughout from outside the room. The experimenter monitored the situation directly outside of the test room to ensure the dog and dyad’s safety and to intervene if needed. The dog owners could terminate the experiment at any point without giving any reason and were permitted to monitor the dog after exiting the room from the video monitor outside, with the experimenter. In cases where any of the dogs experienced extensive stress, the experiment would have been stopped by the experimenter independently of the dog owner; however, this was never the case. As judged by behavioral parameters, none of the dogs experienced stress levels outside of ordinary levels, so in no case did the owner or experimenter intervene to terminate the test situation. All participants were asked to sign consent forms and were informed about their voluntary participation and their right to stop the experiment at any time without providing a reason (see Section 2.11).

### 2.5. Dog Attachment Classification 

The dog’s attachment classification, based on the adapted ASSP, was analyzed according to methods detailed in Schöberl et al. and Solomon et al. [30,34]. Based on the video recordings of the ASSP, coauthors AB and JS, two psychologists trained in attachment categorization of human toddlers, together classified the dogs (with 89% interrater reliability) into five categories: secure, insecure avoidant, insecure ambivalent, insecure disorganized, and unclassifiable. KK and IS assisted with their expertise in dog behavior. Sample sizes and criteria were as follows:

A total of 27 out of 53 dogs were classified as ‘securely attached’: They eagerly approached their owners during the reunion and actively searched for and tolerated physical contact, while also showing interest in exploring the room. Three dogs were classified as ‘insecure-avoidant’, showing little tendency to approach their owner during the reunion but spending much time exploring the room and the toys. Six dogs were classified as ‘insecure-ambivalent’ based on their tendency to not explore their environment after the reunion at all but instead remaining in close proximity to their owner. Nine dogs were categorized as ‘insecure-disorganized’ who showed odd behavioral elements not being part of the normal attachment repertoire, such as freezing, staring or evident stereotypies. For eight dogs, no consensus could be reached; they were therefore labelled ‘unclassifiable’.

### 2.6. Owner Caregiving Rating

The owner caregiving rating was developed by Solomon et al. [34] in the context of previous studies [30,31] and is based on Ainsworth’s maternal sensitivity scale [20] and the ‘supportive presence’ scale [35]. The scale was designed to capture the caregiver’s responsiveness and sensitivity to the dog’s needs in a threatening situation [34]. During the threat task, an unfamiliar person entered the test room wearing a black coat, hat, and ski mask (with only the eyes visible). The unfamiliar person took three steps (in an interval of three seconds) towards the tethered dog while staring at its face. This process was repeated twice. After the second encounter, the unfamiliar person de-escalated the threat by stepping back, taking off the disguise, talking to the dog in a calming matter and offering cheese. The dog owner was present in only one encounter with the unfamiliar threat. The order of the owner’s presence or absence during the encounters was randomized for each dyad. The experimenter (and in one test scenario, also the owner) observed both conditions on monitors outside the room. The dyad participated in the mild threat task four weeks to a year prior to entering the ASS procedure. Caregiving behavior by owners was rated based on the threat task on a seven-point caregiving scale for all dyads. The highest score of seven was given if the owner showed a consistent, quick, and flexible response in their caregiving behavior. The minimum score of one was given if the dog owner did not respond at all to the dog, or if they responded in a negative or punishing way. The ethics of the caregiving scales development was reviewed by the ‘Faculty of Life Sciences’ at the University of Vienna (case number: 2014-015).

### 2.7. Audio Recording

The dog owners’ vocalizations were recorded during the entire ASSP with an H4N recorder connected to a small Sennheiser (ew 100 G3) microphone attached to the clothing in the chest area. The sampling frequency was 48 kHz with 16-bit quantization, and the sensitivity was adjusted prior to recording to prevent clipping. The recording was stopped by the experimenter shortly after entering the room after episode ten.

### 2.8. Audio Treatment and Analysis

All 53 recordings had an adequate signal-to-noise ratio for audio analysis. The audio files of all dyads were prepared for semi-automated, acoustic analysis by hand editing with Audacity (Version 2.1.1; www.audacityteam.org), removing ASSP episodes and episode parts where either a talking second person was in the test room with the owner, or where the dog owner quietly waited outside of the test room (Figure 1). Following this procedure, episodes 1, 4, 7, and 8 as well as minutes two to three of episode 3 were excluded with Audacity (1: experimenter talking in room; 3: strange person talking in room; 4, 7, and 8: owner waiting outside). The remaining episodes 0, 2, 3, 5, 6, 9, and 10 were run through a semi-automated analysis pipeline in Praat (Version 6.0.23; www.fon.hum.uva.nl/praat/). All analysis was conducted using five custom written Praat scripts.

**I. A:** Normalization and prefiltering: Each audio file was adjusted to a maximum amplitude peak of 0.99 and treated with a spectral subtraction background noise filter using Praat commands ‘Scale peak’ and ‘Remove noise’.

**I. B:** Speech segmentation: Praat’s ‘To TextGrid (silences)’ was used to label those portions of the audio containing the owners’ speech. The thresholds for the speech stream segmentation were based on silence in between intervals with a threshold of −35 dB silence, a minimum duration of 0.5 s for silent intervals, and a minimum duration of 0.07 s for spoken intervals. This process labelled the relevant spoken intervals automatically by marking them in the Praat TextGrid. The analyst then verified these automatic labels and made manual accuracy adjustments if necessary, in case of bursts of sound created by the dog vocalizing or playing with the toys.

**II:** Pitch contour tracking: All spoken intervals were saved, and their pitch contours were tracked and extracted using the Praat command ‘Extract visible pitch contour’. The Praat internal commands ‘Get mean’, ‘Get minimum’, and ‘Get maximum’ were used to measure the acoustic parameters mean, minimum, and maximum of each individual pitch contour. 

**III:** Episode matching: A TextGrid tier with episode identifiers was added to match each spoken interval with the corresponding episode. The start and end time of all spoken intervals were extracted.

**IV:** Jitter measurements: The command ‘To PointProcess’ was used to create a point process object out of the pitch contour extracted in script II. Those ‘PointProcesses’ and the command ‘Get jitter (local)’ were used to measure voice’s jitter. The default settings were used except for the ‘Longest period (s)’, where 0.033 (minimum frequency measured) was input.

### 2.9. Analyzed Parameters

Four summary parameters of the f0 (‘voice pitch’) track were used to describe the owner’s speech in spoken intervals throughout the test procedure (Figure 2): voiced duration, mean, range, and jitter. The semiautomated system measured each speech parameter below for every spoken interval. Intervals labelled as spoken but without measurable f0 were automatically excluded from further analysis. For readability, the fundamental frequency (f0) is referred to as ‘pitch’ throughout the rest of the paper.

**Duration:** Voiced utterance duration measured in seconds for each interval and calculated by subtracting the start time from the end time.

**Mean Pitch:** Mean f0 of the spoken interval in hertz.

**Pitch Range:** The minimum f0 measured within the interval, subtracted from the f0 maximum. 

**Jitter:** Jitter was measured using Praat’s algorithm ‘To Jitter (local)’. This parameter is used as a measure of voice quality and is ‘the average absolute difference between consecutive periods, divided by the average period’ (Version 6.0.23; http://www.fon.hum.uva.nl/praat/manual/PointProcess__Get_jitter__local____.html).

### 2.10. Statistical Analysis

Statistical analyses were done using R (Version 3.3.3; www.r-project.org/) and R-Studio (Version 1.0.13; www.rstudio.com/) and the packages ggthemes, ggplot2, psych, data.table, visreg, dplyr, doBy, xlsx, usdm, tidyverse, G.Gally, stats, lme4, car, nlme, lme4, cowplot, multcomp, MuMIn, piecewiseSEM, sjstats, and MASS. The data set for this study is available in the Appendix A. Of all 53 dyads included in this study, over 4100 measurements were analyzed for each of the four response variables. Prior to running detailed analysis, all fixed factors tested negatively for multicollinearity. After visual inspection of the residuals of each response variable, the basic assumption of linear mixed models that the residuals follow a normal distribution could not be confirmed for jitter, mean pitch, pitch range, and voiced duration. Therefore, generalized linear mixed models (glmm) fit using maximum likelihood were calculated for all these four variables. The best distribution fit for each individual model’s response variable was established by investigating the residuals’ histogram and plotting the full models fitted residuals over the estimated residuals. Since the four response variables jitter, mean pitch, pitch range, and voiced duration were continuous and positively skewed towards the right, a Gamma distribution with a log link was used. Dyad was included as random effect to control for individual variances in each model, and the centered and scaled NEO FFI character traits (agreeableness, conscientiousness, extraversion, openness, and neuroticism), centered and scaled caregiving rating, attachment pattern, and owner gender were added as fixed factors and episode as a covariate of the full model. Because voice pitch varies by roughly an octave between adult men and women, gender was highly important to include as a fixed factor for each model. All attachment categories, except for the secure group, were collapsed into the nonsecure attachment group. This was a necessary simplification to reduce the number of levels within categories and facilitate model convergence. Null models included the covariate episode and the random effect dyad. All full models, except the duration model, tested to be a significantly better fit than the null model. Based on the results of the full-null model testing, duration was thus excluded from further analysis. Up to this point, standard practice of fitting linear mixed models was followed. Further decisions for the statistical analysis require some explanation.

A widely-used traditional but criticized approach for finding the best fitting model is a method called stepwise model reduction [36]. This process excludes one variable at a time from the full model based on *p*-values or AICc scores until reaching the null model. Apart from the problems arising from the usage of *p*-values, another issue arises in this context [37]: This stepwise reduction prevents an overview of all possible combinations of fixed factors explaining the response variable. To circumvent the problems and limitations arising through stepwise reduction, we used the function ‘dredge’ of the MuMIn package. This function creates every single model possible out of the fixed factors of the full model with (in our case) AICc scores. The model with the lowest AICc score is picked to provide a baseline, and within a delta of 2 upwards, all models are considered mathematically equally good fits [38]. The models within this range are put through model averaging to create averaged coefficients. This process allows evaluation of the influence all fixed factors have on the response variable without the restriction of *p*-values. For compatibility with the described approach of model averaging based on AICc values, a confidence interval of 85% is used to judge each factor’s impact on the speech parameters [39]. Factors with a confidence interval not including 0 were chosen to be of importance in predicting voice parameters. Relative importance, a measure describing each factor’s relative importance compared to the most valuable factor within the averaged coefficients, is used as a second measure and confirmation of the confidence intervals.

This approach of comparing all models within the delta 2 of the lowest AICc through model averaging was used to gain insight into the complex framework of CADS and its interactions with human personality and the human–dog attachment/caregiving system, without the unnecessary restriction of eliminating valuable comparisons from step one. For a review on null hypotheses significance testing and information theory based approaches and their possible combinations as used here, see Mundry 2011 [40].

### 2.11. Ethics

All participants were asked to sign consent forms, were informed about the procedure, and could terminate participation at any time. The ethics regarding human participation in the ASS procedure was reviewed and approved by the German Society for Psychology (Deutsche Gesellschaft für Psycholgie, AB 07_2011). All human/animal data collection and analysis was done in accordance with the declaration of Helsinki and the EU Directive 2010/63/EU for animal experiments. The ethics for this study was reviewed and approved by the ‘Faculty of Life Sciences’ animal welfare committee at the University of Vienna (case number: 2014-015).

## 3. Results

### 3.1. Mean Pitch

We found owner gender to affect pitch across episodes (Table 1). CADS was consistently higher in voice pitch than read speech (male mean: 121 ± 20 Hz SD; female mean: 203 ± 33 Hz SD) or conversational speech (male mean: 119 ± 19 Hz SD; female mean: 204 ± 38 Hz SD; Figure 3). Both male and female owners’ highest median pitch was recorded during the call episodes (male mean: 176 ± 32 Hz SD; female mean: 300 ± 57 Hz SD). In both genders, the reunions, exploration, and encounter had a similar mean pitch. No effect of personality traits on CADS mean pitch was observed for male and female owners.

### 3.2. Pitch Range

We found gender and openness to affect pitch range across episodes (Table 2). Pitch range in CADS and conversation was reduced relative to reading. Both male and female owners showed the highest median pitch range in the reading control condition (Figure 4A). The median pitch range in the reunions, exploration, and encounter episodes in both men and women was lower than the speaking (male mean: 73 ± 48 Hz SD; female mean: 124 ± 84 Hz SD; Figure 4A) and reading controls (male mean: 84 ± 42 Hz SD; female mean: 146 ± 54 Hz SD; Figure 4A). Female but not male owners high in openness showed an increased frequency range (Figure 4B).

### 3.3. Jitter

Jitter results in the CADS condition were highly variable. We found owner gender and openness to influence jitter across episodes (Table 3). The median jitter was lowest in the encounter (male mean: 0.014 ± 0. 009 % SD; female mean: 0.01 ± 0.007 % SD) and the call episodes (male mean: 0.009 ± 0.004 % SD; female mean: 0.007 ± 0.003 % SD) in both male and female owners. The speaking control condition had the highest median percentage of jitter in both men (male mean: 0.018 ± 0.008 % SD) and women (female mean: 0.012 ± 0.005 % SD). Male owners’ reunion and exploration episodes had a lower percentage of voice jitter than their speaking control condition; women’s percentage of jitter in the same episodes had similar levels to the speaking control (Figure 5A). Male owners’ openness values scaled positively with the percentage of jitter in their vocalizations; this effect was comparatively weaker in female owners (Figure 5B).

## 4. Discussion

We analyzed vocal parameters in dog-directed speech during a series of controlled encounters in dog–human dyads and compared these to normal speech (a read passage, or between-human conversation). The staged encounters were designed to elicit arousal in the dogs and caregiving from their owners. In general, we found that CADS was higher in f0 (“voice pitch”) than normal speech but showed a narrower pitch range. Results concerning pitch perturbation (jitter) were quite variable and showed no clear effect of arousal; the most pronounced effect was a considerable decrease in jitter associated with the owner calling the dog’s name in the two “call” episodes. 

Compared to earlier work on CADS, our results support the findings of an increased pitch but fail to support the previously described broader pitch range. Jeannin et al. [29] partially used an approach similar to the ASSP and found voice pitch in dog-directed speech to increase in reunion episodes compared to adult-directed speech. Our results showed the exact same pattern, with a strong increase in voice pitch during the reunion episodes. Gergely et al. [28] reported a broader frequency range in CADS in mothers compared to fathers. We found the same effect in women to using a broader frequency range during CADS than men did. A well-described phenomenon in IDS and CADS is a differential use of voice pitch and frequency range. Pitch is said to communicate affect, whereas the range is used more in attention getting [10,28,41,42,43]. This different function of pitch and range might also be the explanation as to why we did not find a broader pitch range. Pitch range might simply be adjusted strongly in accordance with context. Namely, if the owner wants to draw the dogs’ attention away from something, a broad pitch range could be used. The opposite would be true if the owner already has the dog’s attention and is trying to soothe and calm it [41]. This would explain why the owners used a broader pitch range and a high pitch while calling their dogs and therefore hoping to draw their attention while also communicating positive intentions. The pitch in exploration, encounter, and the reunion episodes was lower in comparison to the call episodes but still increased compared to the read and adult-directed speech. This elevated pitch coincided with the narrowest pitch ranges in the same episodes. Exploration, encounter, and both reunions share in common that most dogs had already directed their attention towards their owner (in a 2013 study, 85% of the dogs followed their owners around in the reunion and encounter episodes [44]); no broad pitch range was needed to keep their focus. The owners continued to use an elevated pitch to communicate positive affect and a nonthreatening situation.

The results regarding jitter were less clear and more variable. Both males’ and females’ adult-directed speech had the highest percentage of jitter compared to the lowest in the call episodes. Female owners’ median jitter slowly returns to the levels of the reading control throughout the exploration, encounter, and reunions. Male owners’ jitter stays at an elevated level during the exploration and reunion episodes. Interestingly, jitter was highest in the adult-directed speech and behaved contrary to what we would have expected. An explanation for this might be found in the hypothesis that an increase in the speakers’ stress leads to a decreased jitter due to higher tension on the vocal folds [1,4]. This might be a parabola-like phenomenon with relaxed and extremely stressed speakers producing the highest jitter values. This hypothesis would explain why we would see jitter to be lowest when the speaker is stressed but not overly so; enough to cause tension (in the body and the vocal folds) yet mild enough to be dealt with. This hypothesis fits best with our results, but further research must be done to empirically assess this claim.

Our work also illustrates the importance of owner gender and owner personality on CADS. Not only does our study support the idea that CADS is gender-specific and variable enough among dyads to constitute an important component of human–dog caregiving and attachment strategy, we also found openness to drive CADS modulation within this ASSP setup. The influence of personality on CADS seems to be gender-specific and may be considered part of gender-specific performance. Men’s openness was positively correlated with utterance jitter, while women showed almost no correlation. Higher scores in openness correlated with a higher pitch range in women but showed no correlation in men. The limited literature regarding personality in the context of CADS restricts us in doing more than speculating as to why openness might influence jitter and pitch range. There might be one intriguing hypothesis explaining the influence of this one personality axis. The ASSP is designed to evoke arousal and a stress response (and therefore cortisol excretion) in the dogs, which in turn causes a similar stress response in owners. Research suggests a link between cortisol stress response, personality, and gender [45,46]. We propose this link to be reflected in the vocal parameters of jitter and pitch range. This possible interaction of personality, cortisol, and CADS might be a potentially interesting field of further research.

## 5. Conclusions

To summarize, our data partially support and partially contradict our initial hypothesis of an increase in voice source variables with increasingly aroused situations. We did find voice pitch to increase in the ASSP setup in comparison to normal speech, but we found the opposite to be true for pitch range. The decrease in pitch range might be caused by the differential use and function of mean pitch and range. With the narrower range, the owners tried to calm the dogs in this stressful setting. Due to limited reports in literature and close to no precedent, our predictions and hypothesis on jitter were less clear. The idea of voice perturbation to increase with arousal was not supported by our data, but it fit the hypothesis of being (at least partially) personality-dependent.

## Figures and Tables

**Figure 1 animals-09-00417-f001:**
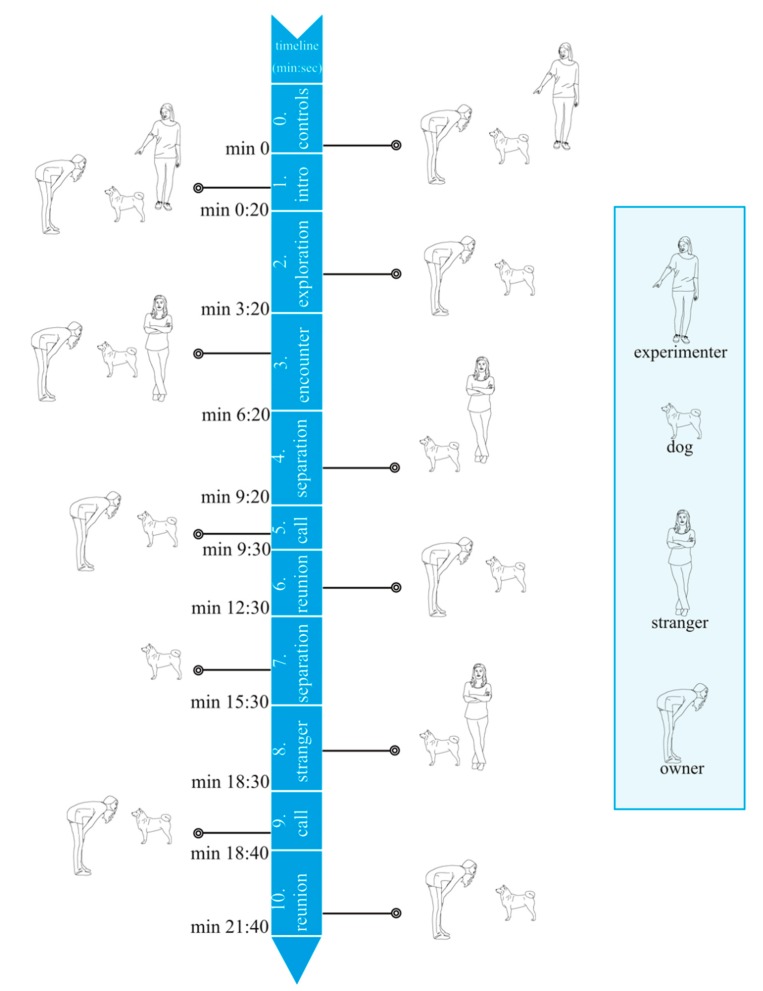
The Ainsworth strange situation (ASS) procedure visualized in the form of a timeline. The icons represent all parties of this study. The dog was present at all times and is therefore marked as such in each episode. The episodes marked with the owner icon were used for audio analysis.

**Figure 2 animals-09-00417-f002:**
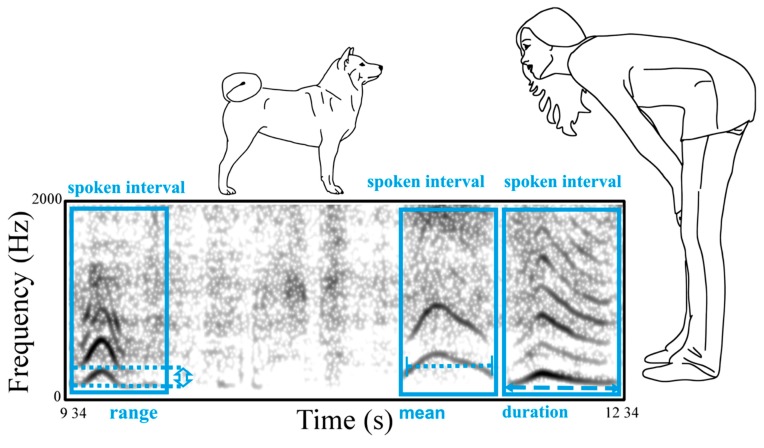
Spectrogram of a female owner talking to her dog during a reunion of the ASS procedure with illustrations of the spoken intervals and the measured variables mean f0 (average f0), voiced duration, and f0 range.

**Figure 3 animals-09-00417-f003:**
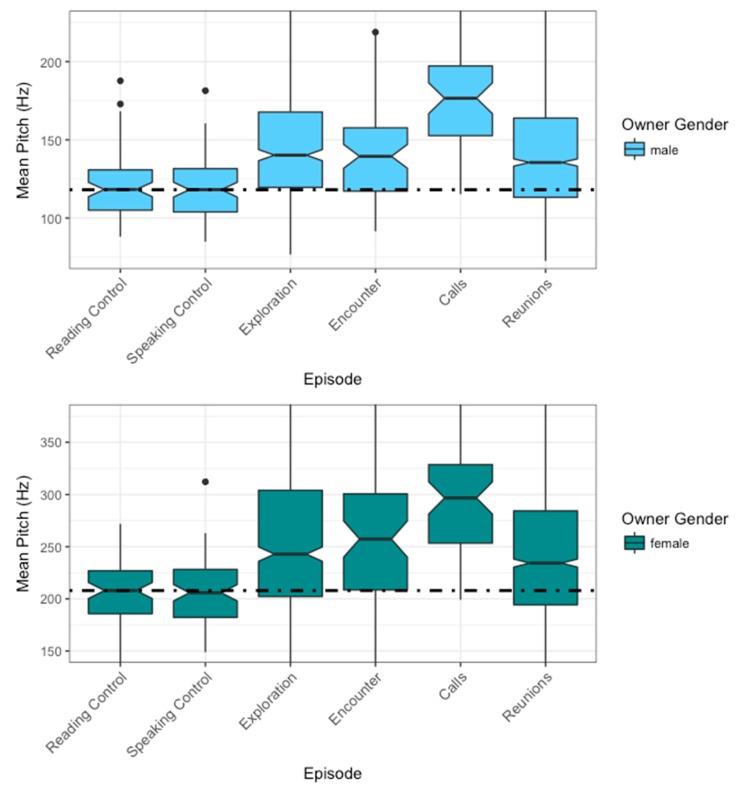
Change in mean pitch by gender over the ASS procedure episodes. The male owners are represented in light blue and the female owners in dark blue. Due to physiological differences, pitch (f0) is separated by about an octave between men and women. Therefore, results are presented in separate graphs (mean pitch plotted by gender over ASS procedure (ASSP) episodes, with the dashed line representing the reading controls median. The two calls and reunions are visualized combined as “calls” and “reunions”.) *n* = 53; number of observations = 4121.

**Figure 4 animals-09-00417-f004:**
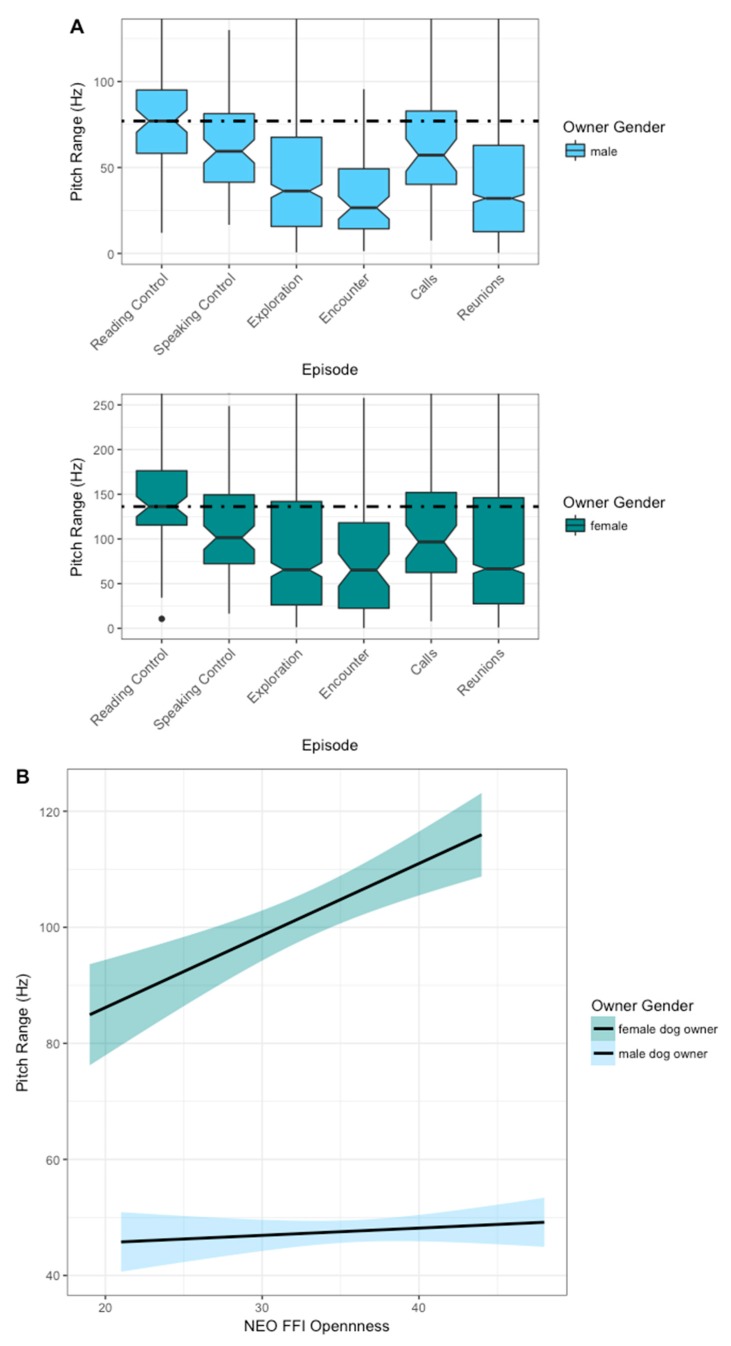
Pitch range by gender over the NEO FFI character trait openness and throughout the ASSP. The light blue line represents male and the dark blue line represents female owner. (**A:** Pitch range plotted by gender over ASSP episodes with the dashed line representing the reading controls median. The two calls and reunions are visualized combined as “calls” and “reunions”. **B:** Pitch range plotted by gender over the NEO FFI openness score.) *n* = 53, number of observations = 4121.

**Figure 5 animals-09-00417-f005:**
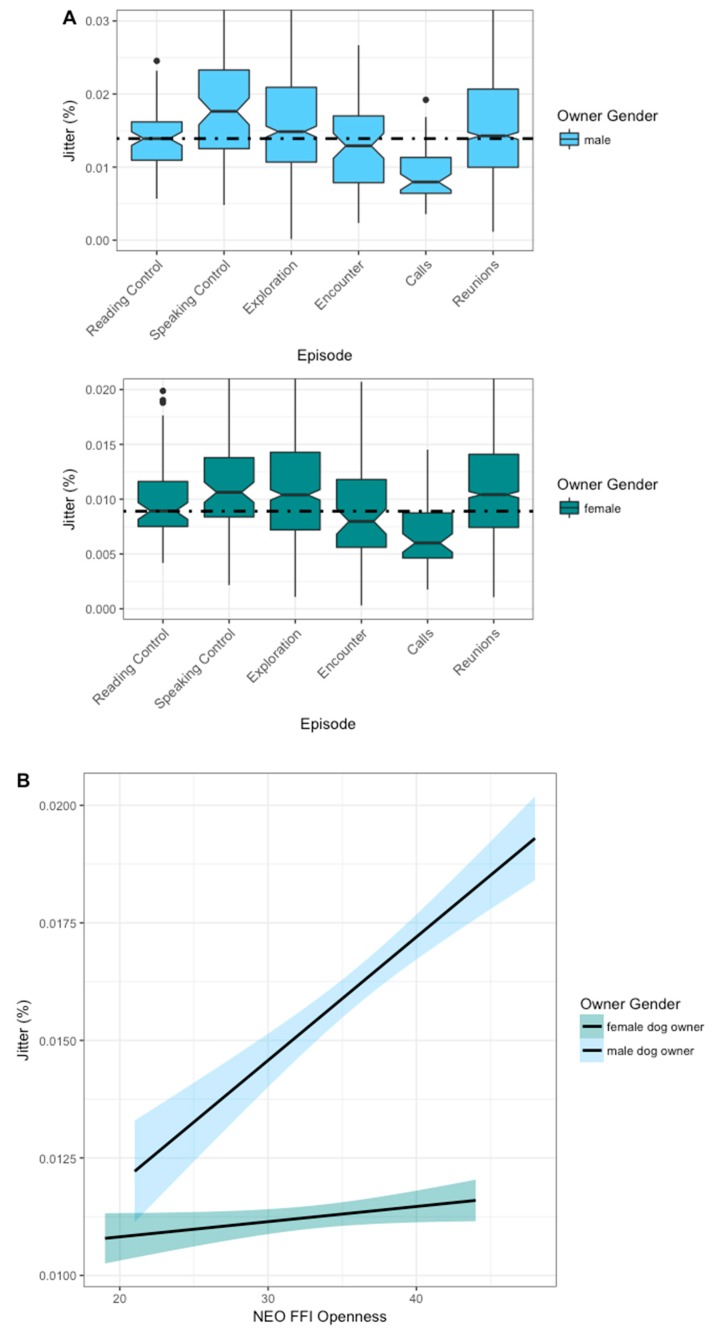
Percentage of jitter by gender over the dog owners’ NEO FFI character trait openness and throughout the ASSP. The light blue line represents male and the dark blue line represents female owner. (**A:** Jitter plotted by gender over ASSP episodes with the dashed line representing the reading controls median. The two calls and reunions are visualized combined as “calls” and “reunions”. **B:** Jitter plotted by gender over the NEO FFI openness score.) *n* = 53; number of observations = 4121.

**Table 1 animals-09-00417-t001:** Model averaged coefficients for mean pitch models. Only variables with a minimum relative importance of 1 and a confidence interval not ranging over 0 are taken into consideration for model interpretation.

Parameter	Estimate	Std. Error	Confidence Interval (85%)		Relative Importance
			0.075	0.925	
(Intercept)	4.7658	0.0429	4.7041	4.8276	
speaking control	−0.005	0.0249	−0.0409	0.0309	1
exploration	0.1904	0.0196	0.1621	0.2187	´´
stranger	0.1859	0.0263	0.148	0.2237	´´
call (1)	0.3863	0.0353	0.3355	0.4371	´´
reunion (1)	0.1822	0.0194	0.1544	0.2101	´´
call (2)	0.3756	0.0348	0.3255	0.4257	´´
reunion (2)	0.1453	0.0197	0.117	0.1736	´´
gender (female)	0.5513	0.0537	0.4739	0.6286	1
caregiving	−0.0027	0.0121	−0.0596	0.0197	0.13
extraversion	0.002	0.0101	−0.0186	0.0506	0.13
attachment (non-secure)	−0.0035	0.0093	−0.0301	−0.0259	0.13
neuroticism	0.0015	0.0106	−0.0281	0.0545	0.12
openness	0.0009	0.0099	−0.033	0.0502	0.11
agreeableness	0.0008	0.0089	−0.0299	0.0451	0.11

Intercept includes reading control, secure attachment, and owner gender male.

**Table 2 animals-09-00417-t002:** Model averaged coefficients for frequency range models. Only variables with a minimum relative importance of >0.8 and a confidence interval not ranging over 0 are taken into consideration for model interpretation.

Parameter	Estimate	Std. Error	Confidence Interval (85%)		Relative Importance
			0.075	0.925	
(Intercept)	4.3426	0.0911	4.2114	4.4738	
speaking control	−0.1656	0.1013	−0.3114	−0.0199	1
exploration	−0.6098	0.0799	−0.7248	−0.4948	´´
stranger	−0.7518	0.1072	−0.9062	−0.5974	´´
call (1)	−0.2962	0.1436	−0.5028	−0.0895	´´
reunion (1)	−0.5974	0.0787	−0.7107	−0.4841	´´
call (2)	−0.3351	0.1415	−0.5388	−0.1315	´´
reunion (2)	−0.618	0.079	−0.7317	−0.5042	´´
gender (female)	0.788	0.0782	0.6755	0.9006	1
openness	0.0613	0.0436	0.0179	0.127	0.85
extraversion	0.0157	0.0303	−0.0016	0.0997	0.32
agreeableness	0.0038	0.0159	−0.0158	0.0858	0.11
neuroticism	0.0059	0.0217	−0.0259	0.0945	0.17
attachment (non-secure)	0.0053	0.0281	−0.0489	0.1668	0.09
conscientiousness	0.0018	0.012	−0.0293	0.0751	0.08
caregiving	−0.0006	0.0099	−0.0624	0.0435	0.07

Intercept includes reading control, secure attachment, and owner gender male.

**Table 3 animals-09-00417-t003:** Model averaged coefficients for jitter models. Only variables with a minimum relative importance of >0.8 and a confidence interval not ranging over 0 are taken into consideration for model interpretation.

Parameter	Estimate	Std. Error	Confidence Interval (85%)		Relative Importance
			0.075	0.925	
(Intercept)	−4.3029	0.0599	−4.389	−4.2167	
speaking control	0.2193	0.0532	0.1427	0.2959	1
exploration	0.1795	0.0418	0.1193	0.2397	´´
stranger	0.0309	0.0563	−0.0501	0.1119	´´
call (1)	−0.4104	0.0754	−0.5189	−0.3019	´´
reunion (1)	0.146	0.0413	0.0865	0.2055	´´
call (2)	−0.2983	0.0746	−0.4057	−0.191	´´
reunion (2)	0.1679	0.0415	0.1082	0.2276	´´
gender (female)	−0.3087	0.0652	−0.4025	−0.2148	1
openness	0.0531	0.0358	0.0156	0.1063	0.87
caregiving	0.0037	0.0145	−0.0181	0.0686	0.15
extraversion	−0.0023	0.0115	−0.0575	0.0212	0.13
agreeableness	0.001	0.01	−0.0331	0.0509	0.11
attachment (non-secure)	−0.0013	0.0207	−0.1025	0.0784	0.1
neuroticism	−0.0004	0.0103	−0.0496	0.0422	0.1

Intercept includes reading control, secure attachment, and owner gender male.

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
