# Peer review of "Talking to Dogs: Companion Animal-Directed Speech in a Stress Test"

_animals, 2019, doi:10.3390/ani9070417_

Round 1
Reviewer 1 Report
Animals-530757
Comments and Suggestions for Authors
General comments: The study deals with Companion Animals Directed Speech (CADS) and the hypothesis that the owner’s gender and personality affect it. To this end, the authors analyzed the parameters of the owners' speech in the context of the Ainsworth Strange Situation Procedure, a test that stimulates the attachment system between subjects (owner and dog) through episodes of separation and reunion.
The paper is well written. Simple Summary and Abstract are clear and easily readable. Materials and Methods are a bit complex to follow, but this is due to the different aspects analyzed in the study. Probably moving paragraphs 2.2, 2.3 and 2.4 further (before paragraph 2.10) could help the reader to know the test procedure before understanding the measures taken for the registration.
Overall, the paper is interesting and well-conceived.
Some changes in the text are required.
Simple Summary:
Line 21: Remove the comma after "conclude" and add it after "results".
Line 23: Remove the second “in”.
Abstract:
Line 31: Add the comma after “Further”.
Line 35: Replace “infant directed” with “infant-directed”.
Introduction:
Line 59: Replace “infant directed” with “infant-directed”.
Line 66: Replace “adult directed” with “adult-directed”.
Materials and Methods:
Lines 92: The number of human-dog teams is wrong. Replace “59” with “53”.
Line 94: The authors stated that dogs are a subset of the 132 human-dog dyads that have participated in other studies. Can they specify the selection criteria of the 53 dyads with respect to the total sample of 132?
Line 98: Remove the comma after “years”.
Line 112: Following the Instruction for Authors, the figures and tables must be indicated in the text as “Figure” and “Table”. Replace "Fig. 2" with "Figure 2". Adapt throughout the text (for example in lines 141, 203 and in the Results).
The numbering of the figures should follow the order in which they appear in the text. Figure 2 appears in the text before the figure 1. The referee invites the authors to change the order of Figures 1 and 2 or to remove the reference to Figure 2 in line 112.
Line 167: Add the comma after “Before testing”.
Line 186: Add the comma after “minute three”.
Line 192: Add the comma after “minute three”.
Line 195: Remove the space between “8” and “Encounter”.
Line 202: Add the comma after “minute three”.
Line 216: Add the comma after “stress”.
Line 231: Replace “59” with “53”.
Line 247: Add the comma after “encounter”.
Line 263: Add the comma after “study”.
Line 266: Add the comma after “variable”.
Line 277: Add the comma after “women”.
Line 278-279: Add the comma after “categories” and “secure group”.
Line 282: Add the comma after “models” and “duration model”.
Line 285: Remove the colon or continue the sentence on the same line.
Line 289: Add the comma after “p values”.
Line 291: Add the comma after “reduction”.
Line 298: Add the comma after “values”.
Line 302: Add the comma after “coefficients”.
Line 305: Add the comma after “system”.
Results:
See previous comment (line 112).
Line 320: Indicate the units of measurement in brackets.
Line 349: There is a mistake. Replace "Figure 1" with "Figure 4".
Lines 353 and 370: Replace “NEO-FFI” with “NEO FFI”.
Line 357: Indicate the units of measurement in brackets.
Line 367: There is a mistake. Replace "Figure 2" with "Figure 5".
Discussion:
Line 408: Add the comma after “Interestingly”.
Line 437: Add the comma after “precedent”.
References:
The references do not follow the journal template. The referee suggests adapting the references according to the Instruction for Authours of the journal (examples shown below):
1. Author 1, A.B.; Author 2, C.D. Title of the article. Abbreviated Journal Name Year, Volume, page range.
2. Author 1, A.; Author 2, B. Title of the chapter. In Book Title, 2nd ed.; Editor 1, A., Editor 2, B., Eds.; Publisher: Publisher Location, Country, 2007; Volume 3, pp. 154–196.
3. Author 1, A.; Author 2, B. Book Title, 3rd ed.; Publisher: Publisher Location, Country, 2008, pp. 154–196.
4. Author 1, A.B.; Author 2, C. Title of Unpublished Work. Abbreviated Journal Name stage of publication
(under review; accepted; in press).
5. Author 1, A.B. (University, City, State, Country); Author 2, C. (Institute, City, State, Country). Personal communication, 2012.
6. Author 1, A.B.; Author 2, C.D.; Author 3, E.F. Title of Presentation. In Title of the Collected Work (if available), Proceedings of the Name of the Conference, Location of Conference, Country, Date of Conference; Editor 1, Editor 2, Eds. (if available); Publisher: City, Country, Year (if available); Abstract Number (optional), Pagination (optional).
7. Author 1, A.B. Title of Thesis. Level of Thesis, Degree-Granting University, Location of University, Date of Completion.
8. Title of Site. Available online: URL (accessed on Day Month Year).
Lines 497 and 508: Use the italics for “Canis familiaris”.
In my opinion, the paper can be accepted for publication after minor revisions.
Reviewer 2 Report
The work is well written and deserves to be published even if being focused mainly on human vocalizations.
I have only a few minor suggestions for the authors.
Introduction:
Please explain better in the introduction the mening of Jitter parameter.
line 44: "presumably rely primarily on such non-verbal information to interpret speaker state and intentions".
The following paper should be cited at this point:” Siniscalchi, M.; d’Ingeo, S.; Fornelli, S.; Quaranta, A. Lateralized behavior and cardiac activity of dogs in response to human emotional vocalizations. Sci. Rep. 2018, 8, 77. doi: 10.1038/s41598-017-18417-4.”
Line 70:” "give and receive this emotional and social support depends on and varies with arousal level and attachment representations".
The following paper in which the arousal level and physiological reactivity of dogs to human emotional faces are discussed should be cited here:”Siniscalchi, M.; d’Ingeo, S.; Quaranta, A. Orienting asymmetries and physiological reactivity in dogs’ response to human emotional faces. Learn. Behav. 2018, 46, 574-585. doi: 10.3758/s13420-018-0325-2”.
Materials adn Methods
It is not clear to me on what basis the authors selected the 59 pairs as subsamples of 132 dyads (line 94).
Discussion
Line 425: “The ASSP is 425 designed to evoke arousal and a stress response (and therefore cortisol excretion) in the dogs which 426 in turn causes a similar stress response in owners”.
The following paper in which a correlation between the owner’s personality and the dog’s
behaviour during the ASSP is reported should be discussed here:” Siniscalchi M, Stipo C, Quaranta A (2013) "Like Owner, Like Dog": Correlation between the Owner’s Attachment Profile and the Owner-Dog Bond. PloS ONE 8(10): e78455. doi:10.1371/journal.pone.0078455”.
